# MobiSim-Bench: A Multi-Perspective Benchmark for Evaluating LLM-Agent-Based Human Mobility Simulation

## Abstract

With advances in large language models (LLMs) and agent technology, LLM agents are transforming social science research on human behavior simulation with their powerful role-playing capabilities. Among the simulation studies on complex human behaviors, mobility behavior simulation has been receiving widespread attention and has important implications for real-world applications. Unlike data-driven statistical learning approaches, LLM agent-based simulation methods have the potential to support all-day simulation and generation of human mobility behaviors or even simulation of adaptive changes in the environment in extraordinary scenarios. To evaluate the performance of LLM agents for human mobility behavior simulation from multiple perspectives and in a holistic manner, we first propose an evaluation framework, which contains three perspectives: **Robustness**, **Realism**, and **Responsiveness**. To implement the evaluation framework, we construct and publish a multi-perspective benchmark named **MobiSim-Bench** based on the AgentSociety simulation framework. The benchmark contains the **Daily Mobility Simulation** mainly for evaluating realism and the **Hurricane Mobility Simulation** mainly for evaluating responsiveness. Based on this benchmark, we organized a challenge with 18 teams to collect and evaluate LLM agents designed by different researchers. In this challenge, 967 agents were deployed. The agent design approach using LLM as the brain achieves the optimum in terms of realism, while the LLM as an extra is more suitable for the responsive scenario. The results show that our evaluation framework and benchmark do examine the performance of LLM agent for simulating human behavior from different perspectives, and on the other hand, they also reveal the shortcomings of the existing LLM agent designs, which will drive the research community to further explore the LLM agent design approaches that can satisfy robustness, realism and responsiveness simultaneously. The benchmark codes are available at `https://anonymous.4open.science/r/MobiSim-Bench-1077/`.

## 1 Introduction

With the development of large language modeling (LLM) (Brown et al., 2020; Touvron et al., 2023; Zhao et al.) and LLM agent technology (Wang et al., 2024b; Fang et al., 2025), LLM agents have not only reshaped the way of working in fields such as programming (Hong et al., 2023a; Yang et al., 2023; Qian et al., 2024), but also changed the paradigm of social science research about the simulation of human behaviors (Park et al., 2023; Gao et al., 2023; Li et al., 2024) with their powerful role-playing capabilities (Shao et al., 2023; Chen et al., 2024). LLM agents inherit the idea of agent-based modeling (ABM) (Schelling, 1971; Deffuant et al., 2000) and replace the agents from manually formulated rules to LLMs that can simulate the logic of complex human behaviors (Gao et al., 2024), which have been successful in the fields of mobility behavior simulation (Shao et al., 2024a; Feng et al., 2024; Yan et al., 2024), social simulation (Gao et al., 2023; Park et al., 2023), economic simulation (Horton, 2023; Li et al., 2024), etc.

Among the simulation studies of complex human behaviors, mobility behavior simulation has been receiving extensive attention (WU et al.; Zhang et al., 2024b; Feng et al., 2024). Accurate simulation of human mobility behavior patterns is of vital importance for urban planning (Neumann et al.,

2019), traffic management (Zhang et al., 2024a), epidemic control (Han et al., 2025), business decisions (Garcia-Gabilondo et al., 2024), etc. From a technical point of view, the LLM agent-based human mobility behavior simulation approach models individual behaviors from the first principle in a way that can overcome the shortcomings of data-driven statistical learning predictive models (Feng et al., 2018; Chen et al., 2020) or generative models (Wang et al., 2021; Yuan et al., 2022) that can only restore the macroscopic distribution. Unlike predictive or generative models, the step-by-step simulation approach (Zhang et al., 2025b) not only embodies the interaction between humans and urban infrastructures, such as road networks, to reflect physical law constraints and thus ensures realism from the microscopic perspective, but also captures the complex intentions and scenario adaptations behind the behaviors through the LLM reasoning process in the simulation. However, existing works (Wang et al., 2024a; Feng et al., 2024) on predicting or simulating human mobility behaviors based on LLM agents still continue the research ideas of statistical learning models, with next-location prediction or trajectory generation as the main research question. They fail to focus on the fact that the LLM agent's role-playing ability with human common sense understanding and reasoning has the potential to support all-day simulation of human mobility behaviors as well as the simulation of adaptive changes in the environment in extraordinary scenarios. Therefore, we believe that long time scale simulation on the day level and different external environment effects are the key to test whether the LLM agents are capable of performing the mobility simulation task and generating highly realistic human mobility behaviors. At the same time, the evaluation of the simulation results should also go deeper from the macro-distribution statistics to the behavioral intention level.

To achieve this, we propose an evaluation framework for comprehensively evaluating the LLM agent's simulated human mobility behaviors from multiple perspectives as follows:

- **Robustness:** First of all, as the most basic requirement, the LLM agents should be able to complete long time-scale mobility simulations for a day or even longer without errors.
- **Realism:** Second, simulation results based on LLM agents should approximate real-world human data in terms of microscopic intentions and macroscopic statistical metrics.
- **Responsiveness:** Unlike data-driven modeling approaches, LLM agent-driven simulation approaches have advanced thinking and reasoning capabilities and should be able to show responsiveness to different external environmental changes.

Based on this evaluation framework, we construct and publish a multi-perspective benchmark named **MobiSim-Bench** to advance the research related to the simulation of human mobility behaviors using LLM agents. MobiSim-Bench consists of two day-level long time-scale simulation tasks, **Daily Mobility Simulation** under normal conditions and **Hurricane Mobility Simulation** under abnormal conditions. To build the two tasks, we collected real-world mobility data, constructed the agent profiles and map data used for initializing agents for the simulation tasks. The real human behaviors were extracted as ground truth for evaluation. In terms of evaluation methods, we not only include common macroscopic statistical distribution metrics, but also further add individual behavioral intention determination. For the hurricane scenario, we also design behavioral change metrics to evaluate the adaptability of the agents. Overall, MobiSim-Bench fully evaluates the realism of LLM-agent-based human mobility simulation in terms of microscopic intentions and macroscopic statistical metrics from both normal and abnormal scenarios, and also examines the environmental adaptability under the impacts of external environmental changes, which realizes a multi-perspective evaluation of long-scale human mobility simulation.

We organized a competition based on our benchmark and collected LLM agents constructed by human experts. A total of 18 teams participated in the competition, submitting 967 agent implementations. We systematically classified all submitted methods into three categories based on the role of LLMs in the agent architecture: LLM as Brain, LLM as Glue, and LLM as Extra. These specially designed agents achieved peak scores of 66.38 in the Daily Mobility task and 85.63 in the Hurricane Mobility task. These competitive outcomes confirm that MobiSim-Bench enables rigorous evaluation of diverse agent design paradigms and validates our multi-dimensional framework for measuring robustness, realism, and responsiveness in long-term mobility simulations.

Overall, the main contributions of this paper are listed as follows:

- We propose an evaluation framework for comprehensively evaluating the LLM agent's simulated human mobility behaviors from robustness, realism, and responsiveness.

- We construct and publish a multi-perspective benchmark named **MobiSim-Bench** with the **Daily Mobility Simulation** task under normal conditions and the **Hurricane Mobility Simulation** task under abnormal conditions for benchmarking.
- We organized a competition to collect, quantitatively evaluate, and compare agents from different teams with different design paradigms in mobility simulation scenarios, and provided corresponding baselines.

## 2 EVALUATION FRAMEWORK

Introducing LLM agents to simulate human behavior will release the high potential of LLMs' role-playing, understanding, and reasoning capabilities in this research field. Unlike prediction or generative algorithms driven by data and statistical learning models, step-by-step simulated LLM agents provide us with a window to simultaneously observe behavioral motivations and behavioral outcomes, while also possessing the potential to respond to environmental changes.

Given these differences, relying solely on statistical distributions of simulated mobility behaviors to evaluate LLM agents has become inadequate. Thus, we propose a hierarchical evaluation framework comprising three key elements as shown in Figure 1 based on the technical characteristics and potential of LLM agents: **Robustness**, **Realism**, and **Responsiveness**. Each element reflects the capabilities required for an LLM agent to simulate human mobility behavior and the corresponding evaluation metrics.

### 2.1 ROBUSTNESS

Robustness is the most fundamental requirement for LLM agents. Unlike statistical learning models that take fully standardized inputs and also produce standardized output matrices, LLM agents directly handle diverse inputs. These inputs include predefined character profiles and external commands, as well as callable functions and even program errors encountered during execution. LLM agents are required to correctly handle all situations and continue simulation with any inputs. This requires the agents to be able to follow instructions to conduct simulations, while also correctly utilizing functions through structured output or function call capabilities. Retry mechanisms and fallbacks will prevent the program from crashing in the event of an LLM error. After ensuring the program simulates normally, the agent's ability to maintain context during day-level long-term simulations

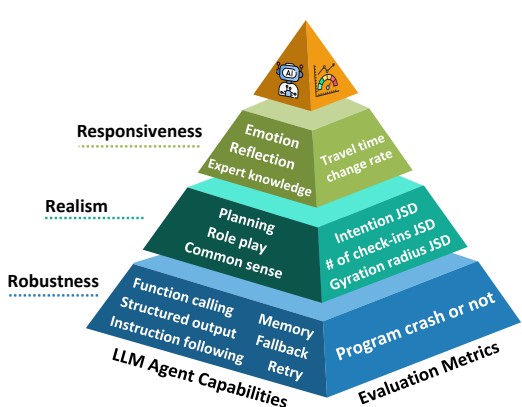

Figure 1: A hierarchical framework for evaluating LLM agents that simulate human mobility behaviors.

will prevent absurd outcomes, such as continuously going out to eat. Memory mechanism (Zhang et al., 2025c) serves as the primary solution to this problem.

Currently, with the advancement of LLM capabilities and the maturation of agent development frameworks (Gao et al., 2025; Zhang et al., 2025a), researchers can easily adopt their built-in mechanisms to build the aforementioned capabilities and create an agent program. Memory systems for LLM agents like mem0 (Chhikara et al., 2025) and A-MEM (Xu et al., 2025) have also significantly reduced the complexity of designing memory modules.

The evaluation of robustness is relatively straightforward: it involves observing whether the agent can complete the simulation without crashing. Any agent that crashes will be vetoed outright.

### 2.2 REALISM

Realism represents a further requirement that is imposed on LLM agents in the task of simulating human mobility behavior. In human mobility behavior simulation tasks, LLM agents are tasked with

playing the role of individuals possessing specific profiles and self-determining their travel choices within a virtual city. This will evaluate the role-playing capabilities of LLM agents, specifically whether they can perform travel behaviors consistent with a given profile, such as a student always commuting to school or a white-collar worker heading to an office building. At the same time, LLMs' ability to understand and apply human societal common sense and routine behavioral patterns, such as sleep schedules, commuting habits, and dining preferences, will also determine whether LLM agents can realistically simulate human mobility. Planning capabilities are also essential when dealing with long-duration simulations to ensure that movement behaviors throughout the day remain relevant and reasonable.

The evaluation of realism can continue to utilize common macro-level statistical distribution metrics from prediction and generation tasks (Feng et al., 2018; Yuan et al., 2022), such as check-in count distribution and gyration radius distribution similarity metrics. Furthermore, since the reasoning processes of LLM agents can be observed by researchers through natural language, we can introduce the recording and evaluation of micro-level behavioral intentions. This will help researchers understand whether LLM agents truly think and act like humans.

### 2.3 RESPONSIVENESS

Responsiveness is the key to surpassing statistical learning models when using LLM agents to simulate human mobility behavior. Statistical learning models rely entirely on the similarity of data distributions to achieve generalization, rendering them useless when facing rare out-of-distribution scenarios. The powerful understanding provided by LLMs to agents will enable these agents to receive and process natural language descriptions of environmental changes in abnormal situations. Leveraging the built-in knowledge and reasoning capabilities of LLMs, they have the ability to transform these descriptions into adaptive actions in response to environmental shifts. During this adaptation process, expert knowledge helps the agent fully comprehend the impacts of external environmental changes and implicitly suggests travel plans. Emotions such as panic and fear reinforce the agent's decision-making preferences, making them more aligned with real-world human behavior. Reflective capabilities primarily focus on whether the agent can re-plan and alter travel arrangements based on factors like external environmental shifts, expert knowledge, and its own emotional state.

For the evaluation of adaptability, we suggest focusing on whether the differences in travel behavior before and after external environmental changes align with real-world conditions. For instance, this could involve computing the similarity in the distribution of travel time changes before and after the changes.

In summary, the proposed framework for assessing robustness, realism, and responsiveness establishes a step-by-step standard for evaluating the performance and potential of LLM agents in simulating human mobility behavior. This framework facilitates the transition of research paradigms in mobility simulation from statistical fitting toward understanding-based behavior modeling, opening new avenues for exploring human movement patterns within complex dynamic environments.

## 3 MOBISIM-BENCH

### 3.1 BENCHMARK OVERVIEW

To implement the proposed evaluation framework to evaluate the robustness, realism, and responsiveness of LLM agents, we introduce a multi-perspective benchmark named **MobiSim-Bench**. The benchmark consists of two tasks: **Daily Mobility Simulation** and **Hurricane Mobility Simulation**. Both tasks are designed to evaluate LLM agents in the domain of human mobility simulation, but under different contextual conditions. The daily mobility simulation is primarily aligned with the goal of **Realism**, as it focuses on capturing routine, everyday urban travel behaviors and assessing whether simulated outputs can approximate real-world human mobility at both microscopic and macroscopic levels. In contrast, the hurricane mobility simulation is closely tied to the goal of **Responsiveness**, as it targets behavioral changes and adaptive responses during extreme weather events, thereby testing whether LLM agents can dynamically adjust to sudden environmental perturbations.

As illustrated in Figure 2, the entire framework consists of three stages: (i) **Data Preparation**, where ground-truth trajectories and user profiles are integrated with urban networks (for Daily Mobility)

Figure 2: Overview of the **MobiSim-Bench**.

or hurricane-related maps and census statistics (for Hurricane Mobility); (ii) **Simulation**, where agents are initialized with demographic and environmental inputs and executed within the built-in mobility engine of **AgentSociety** (Piao et al., 2025b; Zhang et al., 2025a); this engine performs first-principles simulations at a temporal resolution of 1 second, allowing agents to move through explicit function calls that translate intentions into concrete actions, thereby constructing complete mobility trajectories; (iii) **Evaluation**, where the generated behaviors are compared against real-world data, with Daily Mobility metrics (gyration radius, location number, intention sequence, intention proportion) assessing *Realism*, and Hurricane Mobility metrics (change rate, temporal distribution of travel) assessing *Responsiveness*.

## 3.2 DAILY MOBILITY SIMULATION

**Task Definition:** The simulation requires each agent to generate temporally ordered intentions and corresponding actions that are consistent with user characteristics and the surrounding urban environment. The *inputs* include user demographic profiles, city-level geographic and transportation information, and daily time constraints. The *outputs* consist of each agent's concrete mobility behaviors, including intention sequences, executed actions, and the resulting movement trajectories.

**Evaluation Metrics:** To evaluate whether LLM-based mobility simulations approximate real-world human behavior, it is essential to quantify the *similarity* between the generated outcomes and empirical distributions. Similarity-based evaluation provides a principled way to measure how closely synthetic trajectories reproduce both individual-level behavioral intentions and population-level statistical regularities, thereby aligning directly with the *Realism* objective of our framework. We adopt the Jensen–Shannon Divergence (JSD) as the core similarity measure (see Appendix B). Lower JSD values indicate higher similarity between simulated and observed distributions.

The benchmark evaluates four specific aspects, corresponding to different levels of behavioral realism:

- **Intention Sequences JSD**: Consistency in the ordering of individual activity types, directly reflecting behavioral intentions.
- **Gyration Radius JSD**: Similarity in spatial dispersion patterns, capturing the aggregate range of mobility.
- **Daily Location Numbers JSD**: Alignment in the number of distinct places visited per day across the population.
- **Intention Proportions JSD**: Balance among different activity categories at the population level.

To provide a holistic assessment, these four metrics are aggregated into a single *Final Score* (Appendix B). The final score rescales similarity into the normalized range $[0, 100]$, enabling direct comparison across models and tasks.

**Data Preparation:** This benchmark builds upon the processed dataset released in (Shao et al., 2024b), which originates from large-scale mobility records provided by Tencent and China Mobile. The dataset integrates two complementary sources of information: (i) fine-grained mobility trajectories of Beijing users, capturing daily location visits and activity intentions, and (ii) user profile attributes that enrich the contextual understanding of individual behavior. These data enable a comprehensive representation of both movement patterns and demographic heterogeneity, forming the foundation for evaluating LLM-based mobility simulations.

Overall, the Daily Mobility Simulation provides a principled framework for measuring the ability of LLM agents to reproduce realistic urban mobility patterns. By aligning closely with the *Realism* dimension of our evaluation framework, it assesses whether simulated trajectories approximate real-world data at both microscopic (individual-level intentions) and macroscopic (population-level statistics) scales, thereby supporting future research in mobility-aware AI systems.

### 3.3 HURRICANE MOBILITY SIMULATION

**Task Definition:** Agents are required to generate user-level mobility patterns that reflect behavioral variations across three temporal phases: pre-hurricane, during-hurricane, and post-hurricane. The inputs include hurricane-related contextual information, user demographic and behavioral features, and explicit temporal phase indicators, while the outputs consist of each agent's concrete mobility behaviors in terms of travel time and location information.

**Evaluation Metrics:** Two major dimensions are used:

- **Change Rate Accuracy (Change Rate Score):** The accuracy of mobility change rates is measured by mean absolute percentage error (MAPE) (see Appendix B).
- **Distribution Similarity (Distribution Score):** Hourly travel distributions are compared using cosine similarity (see Appendix B).

The final score is a weighted combination of the two metrics (see Appendix B). The weighting emphasizes change rate accuracy (60%) due to its direct reflection of hurricane impact, while distribution similarity (40%) captures temporal mobility dynamics.

**Data preparation:** For the Hurricane Mobility Simulation, we use mobility records obtained from SafeGraph, filtered to include users located in the city of Columbia during Hurricane Dorian. The original SafeGraph data are provided at a weekly resolution, which we further processed into daily trajectories to capture finer-grained temporal dynamics across the pre-hurricane, during-hurricane, and post-hurricane phases. In addition to mobility traces, synthetic user profiles are constructed through a CBG (Census Block Group)-based sampling procedure: (i) identifying the set of CBGs belonging to Columbia, (ii) allocating population samples proportional to each CBG's demographic weight, (iii) assigning residential locations within the sampled CBGs, and (iv) sampling additional attributes such as gender, race, and income level according to CBG-level statistics. This combination of processed mobility trajectories and sampled profile attributes provides a realistic and demographically grounded dataset for evaluating agent responsiveness under extreme conditions.

Overall, the Hurricane Mobility Simulation provides a rigorous and domain-specific framework to evaluate whether LLM agents can replicate mobility adaptations under extreme weather. By emphasizing both change rate fidelity and temporal distribution alignment, it directly reflects the *Responsiveness* dimension of our evaluation framework, testing whether agents can dynamically adjust their behaviors to sudden environmental perturbations and advancing the study of AI-driven human mobility modeling in disaster scenarios.

## 4 EXPERIMENTS

Based on our MobiSim-Bench benchmark, we organized an open competition to evaluate LLM agents under real-world mobility scenarios. A total of 18 teams participated in the competition, with 10 teams entering the daily mobility simulation task and 8 teams entering the hurricane mobility simulation task. Across all submissions, a total of 967 agents were deployed, of which 933 passed the robustness evaluation and obtained valid evaluation scores. Among these, 361 agents were submitted for the daily mobility task and 572 agents for the hurricane mobility task, reflecting both the scale and diversity of approaches explored by the participating teams.

Table 1: Performance of all teams' final submitted agents on the daily mobility task. Boldface indicates best performance.

| Team | Role of LLM | Base Model | $JSD_{gyr}$ | $JSD_{loc}$ | $JSD_{seq}$ | $JSD_{prop}$ | Final Score |
|------|-------------|------------|-------------|-------------|-------------|--------------|-------------|
| #01 | Brain | GLM-4-Flash | 0.328 | 0.665 | **0.063** | 0.289 | **66.38** |
| #02 | Brain | GLM-4-Flash | 0.334 | 0.554 | 0.183 | 0.404 | 63.13 |
| #03 | Brain | GLM-4-Flash | **0.321** | 0.692 | 0.320 | **0.190** | 61.93 |
| #04 | Glue | Qwen-plus | 0.421 | **0.495** | 0.266 | 0.366 | 61.29 |
| #05 | Glue | GLM-4-Flash | 0.329 | 0.655 | 0.170 | 0.404 | 61.04 |
| #06 | Brain | GLM-4-Flash | 0.339 | 0.560 | 0.262 | 0.408 | 60.79 |
| #07 | Extra | GLM-4-Flash | 0.384 | 0.786 | 0.267 | 0.217 | 58.62 |
| #08 | Glue | deepseek-chat | 0.397 | 0.735 | 0.198 | 0.378 | 57.32 |
| #09 | Extra | GPT-4 | 0.433 | 0.720 | 0.253 | 0.522 | 51.80 |
| #10 | Extra | Qwen-plus | 0.393 | 0.791 | 0.639 | 0.441 | 43.39 |

Table 2: Performance of all teams' final submitted agents on the hurricane mobility task. GC represents the Generated Change (During/After vs Before), CE represents the Change Error (During/After vs Before), CRS represents the Change Rate Score, DS represents the Distribution Score. The ground-truth Real Change is -47.34 / -11.50 (During/After vs Before), which is used to compute the Change Error from Generated Change. Boldface marks the best performance.

| Team | Role of LLM | Base Model | GC | CE | CRS | DS | Final Score |
|------|-------------|------------|-----|-----|-----|-----|-------------|
| #11 | Extra | GLM-4-Flash | -44.01 / -12.76 | 3.33 / 1.26 | 91.02 | 77.53 | **85.63** |
| #12 | Extra | deepseek-chat | -45.54 / -9.90 | **1.80 / 1.60** | **91.13** | 64.47 | 80.47 |
| #13 | Brain | GLM-4-Flash | -86.25 / -11.54 | 38.91 / 0.04 | 58.74 | 76.37 | 65.79 |
| #14 | Glue | deepseek-chat | -8.37 / -14.44 | 38.97 / 2.94 | 46.08 | **86.33** | 62.18 |
| #15 | Brain | GLM-4-Flash | -43.30 / -14.43 | 4.04 / 2.93 | 83.00 | 27.56 | 60.83 |
| #16 | Brain | GLM-4-Flash | -41.20 / -66.56 | 6.14 / 55.06 | 0.00 | 85.78 | 34.31 |
| #17 | Brain | GLM-4-Flash | -28.35 / -46.63 | 18.99 / 35.12 | 0.00 | 83.40 | 33.36 |
| #18 | Extra | GLM-4-Flash | -60.05 / -94.75 | 12.71 / 83.25 | 0.00 | 58.01 | 23.20 |

## 4.1 BASELINES

The participating teams adopted a wide range of strategies, and some teams even experimented with multiple approaches. Across all these submissions, we identified three predominant design paradigms, each with distinct roles for LLMs in the decision-making process.

**LLM as Brain:** In this paradigm, teams primarily leverage LLMs' comprehensive reasoning and creative generation capabilities to drive complete agent behavior. These approaches utilize the LLM's natural language processing and multi-step reasoning to produce complex behavioral decisions. The best-performing LLM-dominated approach in the daily mobility task employs a narrative-driven methodology. First, the LLM generates a detailed, first-person daily narrative describing the character's activities and thoughts throughout the day (An example can be found in Appendix A.3). Second, another LLM call parses this narrative into structured activity plans. This two-step process ensures both narrative coherence and structural precision, allowing the agent to follow a pre-generated plan throughout the task day.

**LLM as Glue:** In this paradigm, teams primarily utilize LLMs' contextual adaptation and intelligent bridging capabilities to enhance rule-based systems. These approaches leverage the LLM's ability to understand complex contexts and provide intelligent recommendations that connect different system components. The best-performing LLM-as-Glue approach employs a multi-phase state manager with explicit normal, hurricane, and post-hurricane phases. The system maintains internal states (fatigue, hunger, emotion, personality) and uses predefined behavioral templates, but the LLM provides intelligent recommendations that consider the agent's current subjective state, personality type, and environmental context. When the primary rule-based system encounters complex situations, the LLM bridges the gap between rigid templates and dynamic contextual needs.

**LLM as Extra:** In this paradigm, teams either minimally utilize LLM capabilities or completely avoid them, with core intelligence residing in well-designed rule systems. When present, teams only utilize the LLM's basic pattern recognition abilities as supplementary tools. The best-performing LLM-as-

Extra approach in the hurricane mobility task employs a sophisticated hour-level probability table system that adapts to different hurricane phases and individual agent characteristics. The system defines three distinct phases with corresponding probability matrices that specify movement likelihoods for each hour of the day. The decision logic is entirely deterministic and rule-based, demonstrating that sophisticated agent behavior can be achieved through well-designed rule frameworks without advanced LLM capabilities.

## 4.2 RESULTS

The performance results across the three design paradigms as Table 1 and Table 2 shows reveal distinct trade-offs and characteristics that align with their underlying design philosophies.

**LLM as Brain:** The LLM-as-Brain approach achieved the highest performance in the daily mobility task, demonstrating its strength in generating natural and diverse behavioral patterns. The superior JSD scores for intention sequences and intention proportions indicate that LLM-driven agents excel at producing coherent and psychologically plausible activity sequences. These sequences not only closely match real human behavior patterns at a macro level but also effectively reconstruct specific micro-behaviors such as morning commutes, lunch breaks, and evening leisure activities. This aligns with the daily mobility benchmark's focus on behavioral realism, where the ability to generate natural activity transitions and maintain appropriate intention distributions is crucial. However, when facing other types of environmental changes, such as hurricane scenarios, the current LLM-as-Brain design struggles, producing unrealistic or overly complex behaviors that fail to replicate the simplified yet accurate human responses required during emergencies.

**LLM as Glue:** The LLM-as-Glue approach achieved balanced performance across both tasks, indicating its versatility in handling diverse scenarios. The moderate JSD scores in daily mobility suggest that while state-driven agents can maintain behavioral coherence, they may struggle with the complexity of parameter tuning and state space management. In the hurricane mobility task, the lower distribution score indicates challenges in accurately modeling temporal patterns during extreme weather events.

**LLM as Extra:** The LLM-as-Extra approach showed remarkable effectiveness in the hurricane mobility task, where predictability and reliability are crucial for emergency scenarios. The excellent change rate score demonstrates that rule-based systems can accurately capture the expected behavioral shifts during extreme weather events, which is essential for the hurricane mobility benchmark's emphasis on change rate accuracy. However, the relatively lower performance in daily mobility suggests that rigid rule structures may limit behavioral diversity and creativity in normal circumstances, as evidenced by higher JSD scores across all metrics.

## 4.3 DISCUSSION

**Cross-task Performance Patterns:** The performance patterns reveal how different LLM roles affect the quality of human behavior simulation across different scenarios. In the daily mobility task, which evaluates the ability to simulate realistic daily human movement patterns, LLM-as-Brain approaches excel by leveraging the model's comprehensive reasoning capabilities to generate natural, contextually appropriate behaviors. The superior JSD scores for intention sequences and proportions demonstrate that LLM-driven agents produce more coherent activity patterns that better match real human behavior, suggesting that complex behavioral modeling benefits from maximal LLM involvement.

However, in the hurricane mobility task, which evaluates the ability to simulate human responses during extreme weather events, LLM-as-Extra approaches (with minimal LLM involvement) outperform those with greater LLM integration. This reveals an important insight: for scenarios with clear behavioral patterns and predictable human responses, well-designed rule systems can more accurately model the expected changes in human mobility behavior, while current LLMs may still struggle to represent the precise behavioral patterns and temporal dynamics required for accurate emergency response simulation.

**Cross-model Performance Patterns:** In the hurricane mobility task, one team applied the same rule-driven pipeline with different base models, revealing clear model-specific trade-offs between fitting the overall change rate and preserving hour-level distribution patterns. (as Table 3 shows). Deepseek-chat(DeepSeek-AI et al., 2024) shows the tightest alignment with targeted change rates,

Table 3: Comparison of three base models from the same team using the same paradigm on the hurricane mobility task. Boldface marks the best performance.

| Model | GC | CE | CRS | DS | Final Score |
|---|---|---|---|---|---|
| GLM-4-Flash-Free | -44.01 / -12.76 | 3.33 / 1.26 | 91.02 | **77.53** | **85.63** |
| Deepseek-chat | -46.76 / -11.51 | **0.58 / 0.01** | **99.36** | 59.87 | 83.57 |
| Qwen-plus | -51.18 / -10.49 | 3.83 / 1.02 | 91.53 | 56.18 | 77.39 |

reflecting strong numerical discipline and instruction-following, but this comes at the cost of flatter, less detailed hourly dynamics. GLM-4-Flash-Free achieves the most balanced performance, keeping change errors low while maintaining richer diurnal structures, which supports its leading overall score. Qwen-plus(Yang et al., 2025), by contrast, lags on both metrics, with larger deviations in change rate and weaker reconstruction of hourly usage, indicating less stable phase calibration. These outcomes suggest a practical guideline: choose numerically disciplined models when aggregate accuracy is critical, and balanced models when both accuracy and realistic hourly patterns matter, avoiding models with inconsistent behaviors across metrics.

## 5 RELATED WORK

### 5.1 MOBILITY SIMULATION

Research on mobility behavior simulation can be broadly divided into two categories. The first category follows traditional deep learning approaches, including classical Markov models (Rendle et al., 2010) and subsequent sequence-modeling techniques such as recurrent neural networks (RNNs) (Lai et al., 2023; Feng et al., 2020) and attention-based architectures (Qin et al., 2022; Hong et al., 2023b). More recent studies employ LLM-driven agents to conduct mobility simulations (Feng et al., 2025; Shao et al., 2024b; Wang et al., 2024c), leveraging the agents' extensive world knowledge, reasoning capabilities, and adaptive decision-making to generate more realistic and dynamic movement patterns.

### 5.2 LLM AGENT SIMULATION BENCHMARK

The use of LLM agents for simulation has attracted growing attention in recent years. A number of studies (Sukiennik et al., 2025; Zhao et al., 2024) have demonstrated the broad societal value of deploying LLM agents in complex simulation settings. Meanwhile, platforms such as AgentSociety (Piao et al., 2025b) and YuLan-OneSim (Wang et al., 2025), along with recent efforts to optimize multi-agent simulation systems (Piao et al., 2025a; Zhang et al., 2025a), have further facilitated large-scale agent-based simulation experiments. Despite these advances, most existing LLM-agent benchmarks remain primarily focused on assessing "tool-like" capabilities (Abdelnabi et al., 2024; Zhu et al., 2025; Xu et al., 2024; Piatti et al., 2024), offering limited evaluation of agents' ability to simulate human behavioral patterns. Our work addresses this gap by introducing a benchmark specifically designed to assess agents' competence in modeling realistic human behaviors, thereby contributing a novel and meaningful perspective to the field.

## 6 CONCLUSION

In this paper, to comprehensively evaluate the performance of LLM agent for human mobility behavior simulation, we propose an evaluation framework containing three perspectives: robustness, realism, and responsiveness. Guided by the evaluation framework, we construct a multi-perspective benchmark named MobiSim-Bench powered by AgentSociety simulation framework, which contains the daily mobility simulation and the hurricane mobility simulation. By organizing a challenge, we evaluated the performance of multiple LLM agent design approaches under this evaluation framework and benchmark. Unfortunately, none of the LLM agent designs can achieve robustness, realism and responsiveness at the same time. This demonstrates the importance and value of MobiSim-Bench on one hand, and reveals the inadequacy of current LLM agent designs for simulating human mobility behavior on the other. We hope that MobiSim-Bench can help the research community to explore and discover LLM agent designs that can effectively and comprehensively simulate human mobility behavior, and thus promote the development of social science research paradigms driven by LLM agents.

ETHICS STATEMENT

This work fully complies with the ICLR Code of Ethics. All datasets used in MobiSim-Bench have undergone strict anonymization and desensitization procedures to ensure that no personally identifiable or sensitive information is retained. The benchmark is designed solely for research purposes, emphasizing transparency, reproducibility, and responsible use. Dataset documentation, simulation procedures, and evaluation guidelines are provided to facilitate safe adoption and avoid potential misuse. No conflicts of interest or external sponsorship influenced the design or outcomes of this work.

REPRODUCIBILITY STATEMENT

We prioritize reproducibility by releasing all necessary resources alongside the paper. The datasets used in MobiSim-Bench, preprocessing scripts, simulation workflow, evaluation metrics, and baseline implementations are included in an anonymized repository linked with the abstract. We provide detailed descriptions of the two tasks in our benchmark framework, the Daily Mobility Simulation (in Subsection 3.2) and the Hurricane Mobility Simulation (in Subsection 3.3). Each Subsection specifies the task definition, the datasets employed, and the corresponding preprocessing steps. In addition, Appendix B lists the complete calculation formulas for all evaluation metrics. Considering the inherent randomness of LLMs, in order to reduce the difficulty of reproduction, we release baseline methods for both tasks, with Section 3 reporting extensive results across different LLM configurations. Appendix A.2 further compiles all benchmark results for reference, while Appendix A.3 presents a detailed output example of the best-performing LLM-driven agent in the Daily Mobility task. These resources collectively ensure that independent researchers can reliably reproduce and extend our findings.

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

# A   APPENDIX

## A.1   THE USE OF LARGE LANGUAGE MODELS

In this work, the use of LLMs is limited to checking for grammatical errors and providing word suggestions.

## A.2   OVERALL SUBMISSION STATUS

Table 4: Summary of competition participation.

| task | # of Teams | # of Agents Submitted | # of Agents Evaluated |
|---|---|---|---|
| Daily mobility | 10 | 362 | 361 |
| Hurricane mobility | 8 | 605 | 572 |
| **Total** | **18** | **967** | **933** |

## A.3   AN EXAMPLE OF NARRATIVE GENERATED FROM THE BEST-PERFORMING LLM-DOMINATED AGENT IN THE DAILY MOBILITY TASK

I woke up around 7:45 AM to the sound of my phone buzzing on the nightstand—same old alarm, same tired feeling. My apartment in Haidian (that's location ID 500063798 for whatever system needs that) is cozy but nothing fancy: IKEA furniture, a desk buried under cables and old coffee mugs, and a window that looks out onto a row of elm trees and the back of a convenience store. I lay there for a few minutes scrolling through WeChat—family group messages from my mom asking if I've eaten properly, a couple memes from my college buddies, and a work Slack ping reminding me about today's sprint review.

By 8:10, I dragged myself out of bed. Quick shower, face wash, toothbrush dance—I keep it simple. Breakfast was instant oatmeal with a banana and black tea, all while half-watching a tech podcast on my laptop. I hate mornings, but at least Beijing's autumn air is finally cooling down. No more sweltering subway rides.

Left the house at 8:50, walked five minutes to Wudaokou Station. Took Line 13 toward Zhichunlu, then switched to Line 10. The commute's predictable—packed cars, people reading novels or napping standing up, someone always blasting music by accident. Got off at Suzhoujie Station (close enough to office ID 500059965), then a ten-minute walk through that narrow street lined with bubble tea shops and repair stalls.

Reached the office around 9:25. Clock in, hang up my jacket, refill my thermos with hot water. Team meeting started at 9:30—we're wrapping up this backend migration, so I gave a quick update on the API refactoring I've been grinding on. Then it was heads-down coding until lunch. Debugged a race condition in the user auth module—stupid typo in a timeout config. Felt dumb, but hey, at least it wasn't in production.

Lunch was at 12:30. Me and three other guys from the team hit that hand-pulled noodle place across the street—*Lanzhou Lamian*, yellow signs, cheap and filling. Beef soup, extra cilantro, chili oil on the side. We talked about everything except work: someone's new apartment, the PS5 price hike, whether China will ever get decent sci-fi movies outside of *The Wandering Earth*. Back to the office by 1:15.

Afternoon was lighter—code review for a junior dev, updated some documentation (ugh), and fixed a minor UI glitch on the internal dashboard. Took a 20-minute break around 4:00 to walk around the block and grab a soy milk and steamed bun from the old lady's cart near the park entrance. Autumn sunlight felt good on my face.

Wrapped up around 6:45. Manager didn't push for overtime, which I appreciated—we're not crunching yet. Left the office by 7:00. Instead of going straight home, swung by the Carrefour Express near Wudaokou Square. Picked up tofu, bok choy, instant ramen (for emergencies), a bottle of soy sauce, and a liter of beer—just one, because I'm not trying to become "that guy" with a fridge full of empties.

Got home around 7:45. Changed into sweatpants, put on some lo-fi beats. Cooked a quick stir-fry with what I had—garlic, tofu, greens—and poured the beer into a glass. Ate while watching an episode of *Line of Duty* dubbed in Mandarin. Not the same without the accents, but it passes the time.

Around 9:30, I opened my notebook and sketched out some ideas for a personal project—a habit-tracking app with minimalist design. Nothing serious, just something to tinker with on weekends. Then browsed Zhihu for half an hour, reading rants about AI replacing programmers (again).

Showered again—sweat from the commute still clinging—and brushed my teeth. Checked my phone one last time: no urgent messages, weather says tomorrow will be partly cloudy. Set the alarm for 7:50. Lights out by 11:20.

As I drifted off, I thought: *Another day, another thousand lines of code nobody will ever see.* But hey—at least the noodles were good.

## B   FORMULAS FOR EVALUATION METRICS

### B.1   DAILY MOBILITY SIMULATION METRICS

The Jensen–Shannon Divergence (JSD) is defined as:

$$\mathrm{JSD}(P \parallel Q) = \tfrac{1}{2}\mathrm{KL}(P \parallel M) + \tfrac{1}{2}\mathrm{KL}(Q \parallel M), \quad M = \tfrac{1}{2}(P + Q),$$

where $P$ and $Q$ denote the probability distributions of generated and real-world data, respectively, and $\mathrm{KL}(\cdot \parallel \cdot)$ is the Kullback–Leibler divergence.

The aggregated *Final Score* is defined as:

$$\text{Final Score} = \left( \frac{(1 - \mathrm{JSD}_{\mathrm{gyr}}) + (1 - \mathrm{JSD}_{\mathrm{loc}}) + (1 - \mathrm{JSD}_{\mathrm{seq}}) + (1 - \mathrm{JSD}_{\mathrm{prop}})}{4} \right) \times 100.$$

### B.2   HURRICANE MOBILITY SIMULATION METRICS

The mean absolute percentage error (MAPE) and the Change Rate Score are given by:

$$\mathrm{MAPE} = \frac{|\text{Real Change Rate} - \text{Generated Change Rate}|}{|\text{Real Change Rate}|} \times 100\%,$$

$$\text{Change Rate Score} = \max\big(0, \ 100 - \text{Average MAPE}\big).$$

The cosine similarity and the Distribution Score are defined as:

$$\text{Cosine Similarity}(A, B) = \frac{A \cdot B}{\|A\| \times \|B\|},$$

$$\text{Distribution Score} = \max\big(0, \text{Average Cosine Similarity} \times 100\big).$$

The weighted final score is:

$$\text{Final Score} = 0.6 \times \text{Change Rate Score} + 0.4 \times \text{Distribution Score}.$$

