# OpenReview forum: "MobiSim-Bench: A Multi-Perspective Benchmark for Evaluating LLM-Agent-Based Human Mobility Simulation"
_ICLR.cc/2026/Conference — Submitted to ICLR 2026_

### Official Review · Reviewer_EiMV · 2025-10-31

**Soundness:** 3
**Presentation:** 3
**Contribution:** 3
**Rating:** 4
**Confidence:** 4

**Summary:**

This paper proposes a three-dimensional evaluation framework encompassing robustness, realism, and responsiveness for assessing LLM agents in human mobility simulation, and constructs the MobiSim-Bench benchmark. Through organizing a competition, the authors collected agent designs from multiple teams and evaluated three design paradigms. The paper addresses a potentially valuable research direction, but it suffers from significant weaknesses in theoretical grounding of the evaluation framework, experimental rigor, and reliability of conclusions.

**Strengths:**

1. Addresses a practically relevant problem of applying LLM agents to human mobility behavior simulation
2. Provides two types of simulation tasks covering both daily and extreme scenarios
3. Collects a relatively large sample of 967 agent implementations for evaluation
4. Releases benchmark code and datasets to facilitate research reproducibility
5. Identifies and discusses three distinct agent design paradigms

**Weaknesses:**

- The claimed hierarchical structure of the evaluation framework lacks theoretical foundation, with unclear relationships among the three dimensions. Why robustness is basic while responsiveness is advanced?
- The benchmark design has narrow scenario coverage, containing only daily and hurricane scenarios, which limits comprehensive assessment of agent capabilities and raises concerns about generalizability
- The competition-based data collection results in experiments lacking rigorous controlled variables, as different teams used different models, prompts, and architectural designs
- The evaluation metrics lack adequate justification, as distribution similarity measures may not genuinely reflect agent cognitive and decision-making capabilities
- Missing systematic comparison experiments with traditional statistical learning models
- The paper claims LLM agents can handle out-of-distribution scenarios but provides only one hurricane case as evidence
- Robustness evaluation is overly simplistic, only checking whether programs crash without examining deeper capabilities like consistency in long-term simulation and memory retention
- Overclaiming. The paper uses vague statements like has potential and may support in multiple places without experimentally validating these capabilities

**Questions:**

1. How is comparability ensured among agents submitted by different teams, and were key variables like model size and API call frequency controlled
2. How were intention sequences annotated, how was annotation reliability verified, and what is the inter-annotator consistency
3. Why did the best LLM as Brain agent score only 66 in the daily task
4. How was the ground truth for change rate in the hurricane scenario determined, and why is a change of -47.34% considered a reasonable reference standard
5. If simple rule-based systems perform better in certain dimensions, what is the necessity of LLM agents
6. Why not include other types of extreme scenarios such as pandemics or traffic control to more comprehensively evaluate responsiveness

---

### Official Review · Reviewer_BbWi · 2025-10-31

**Soundness:** 2
**Presentation:** 3
**Contribution:** 1
**Rating:** 2
**Confidence:** 4

**Summary:**

In this paper, the authors propose a benchmark called MobiSim-Bench for evaluating large language model (LLM) agents in human mobility simulation. The benchmark comprises two tasks: simulating daily human mobility and simulating mobility during a hurricane. The evaluation metrics are based on several Jensen–Shannon Divergence (JSD) similarity measures, each designed to capture different properties of the trajectories compared with ground-truth data. Experiments are conducted through an open competition in which teams develop and apply LLM-based strategies.

**Strengths:**

This work emphasizes the importance of modeling human mobility and highlights several key aspects of applying LLMs to this task. The presentation is generally clear and well structured. Developing a benchmark to assess the ability of LLM agents to model human mobility represents a relevant research direction. For example, studying mobility in disaster scenarios could contribute to saving lives.

The benchmark is anchored in real-world ground-truth data, which strengthens its empirical foundation. The use of Jensen–Shannon Divergence (JSD) similarity metrics to compare simulated and real mobility trajectories adds methodological soundness. The public availability of the code is also a positive aspect.

**Weaknesses:**

As a benchmark paper, this work falls short in several aspects expected for a rigorous and well-justified benchmark study.

Unclear rationale:
Although the paper is generally well written and highlights important issues in benchmarking LLM agents for human mobility modeling, the connection between the conceptual discussion and the concrete benchmark design remains unclear. The paper spends considerable effort explaining why benchmarking such agents is meaningful and what makes such benchmarks useful, yet it is difficult to see how these discussions inform specific design decisions in the benchmark. This disconnect weakens the overall rationale and coherence of the work.

Limited justification and unclear novelty:
The motivation for introducing a new benchmark is not sufficiently justified. It remains unclear how the proposed benchmark addresses the limitations of evaluation methods in prior work on LLM-based human mobility modeling, which already define their own evaluation metrics. For instance, AgentMove https://arxiv.org/abs/2408.13986  evaluates models using prediction accuracy, and Large Language Models as Urban Residents https://arxiv.org/abs/2402.14744 measures trajectory distance. Furthermore, Human Mobility Modeling with Limited Information via Large Language Models https://arxiv.org/abs/2409.17495 (a work missed in citation) already applies JSD to assess mobility similarity, using the same metric adopted in this paper. Of course, in a benchmark paper, adapting existing metrics or innovating new ones is entirely reasonable. However, the paper should explicitly discuss how its chosen metrics and setup differ from or improve upon prior efforts, to support the claims of novelty and usability.

Ambiguity in the competition-based evaluation:
The decision to evaluate the benchmark through an open competition is an interesting and bold choice (this is a good bold move per se). However, this setup introduces ambiguity about what is being measured. It is difficult to disentangle (probably due to missing details) the contribution of human-designed strategies from the inherent capabilities of the LLM agents, which poses a problem for a benchmark intended to evaluate the agents themselves. Both aspects could independently merit study, yet the manuscript provides limited detail on how team strategies were formulated or controlled. As a result, it is hard to interpret the results and assess the benchmarks’ reliability.  The lack of interpretability analysis (e.g. what the models do well and where they fail in the benchmark) adds more confusion in this aspect.

**Questions:**

(See the missing parts in weakness)

---

### Official Review · Reviewer_4Nb8 · 2025-11-01

**Soundness:** 2
**Presentation:** 1
**Contribution:** 2
**Rating:** 2
**Confidence:** 5

**Summary:**

This paper proposes a framework for evaluating the performance of LLM agents for human mobility generation, from the perspective of robustness, realism, and responsiveness. Two tasks are involved in the framework, one for daily mobility and the other for an extraordinary scenario of a hurricane. The evaluation is conducted over 18 teams of 967 LLM agents, with different models in three design paradigms evaluated.

**Strengths:**

S1. The use of LLM agents for mobility simulation is a promising direction.

S2. The evaluation framework is comprehensive in the sense that it covers three perspectives: robustness, realism, and responsiveness.

S3. Multiple LLMs are evaluated in the simulations.

**Weaknesses:**

W1. The positioning of the paper is unclear and overclaimed. In the introduction, the paper claimed existing works (Wang et al., 2024a; Feng et al., 2024) "fail to focus on the fact that the LLM agent's role-playing ability with human common sense understanding and reasoning has the potential to support all-day simulation of human mobility behaviors as well as the simulation of adaptive changes in the environment in extraordinary scenarios". This is exactly what was done in Wang et al., 2024a, though another extraordinary scenario of a pandemic is evaluated instead of a hurricane. In addition, Wang et al., 2024a involves the design of intentions, which are translated into actions.

W2. The presentation needs to be improved. Too many details are missing. For example, the descriptions of the three agent design paradigms are too vague. Only an example for the LLM as Brain setting on daily mobility simulation is given in the appendix, and this is the output rather than the input. Rule-based systems are mentioned in the LLM as Glue and LLM as Extra settings, but it is completely unknown how these rules-based systems work.

W3. It is unclear how the agent teams are implemented. Do team members have any interactions?

W4. The analysis of the simultion results is too brief. The results show that LLM as Brain excels in daily mobility but turns out to be less competitive and LLM as Extra in hurricane mobility. This is interesting, but the paper does not delve into detailed analysis. A win/loss comparison or failure mode analysis may help understand the takeaways of simulation results.

**Questions:**

Q1. Are the generated trajectories on the GPS level or POI level? I suppose they are on the GPS level, so you can measure the gyration radius.

Q2. How long are the generated trajectories, in particular, the range and the average length of a trajectory?

Q3. How are the agents instructed to perform both tasks?

Q4. Do you use any few-shot prompts or ground truth data to calibrate the behavior of agents?

Q5. In Table 1, Teams #01-#03 all adopt LLM as Brain and use GLM-4-Flash as the base model. What's the difference between them?

Q6. Table 3 shows the cross-model performance patterns, but why are they only evaluated for the hurricane mobility task?

---

### Official Review · Reviewer_F3dw · 2025-11-02

**Soundness:** 2
**Presentation:** 1
**Contribution:** 2
**Rating:** 2
**Confidence:** 3

**Summary:**

The paper proposes MobiSim-Bench, a benchmark and evaluation framework for LLM-agent mobility simulation along three axes: Robustness, Realism, and Responsiveness. It instantiates two day-level tasks: Daily Mobility (Beijing mobility data with intention annotations) and Hurricane Mobility (Columbia users during Hurricane Dorian from SafeGraph). Metrics combine macro distributions (e.g., gyration radius, #locations) and micro intentions via JSD for Daily, and change-rate accuracy + hourly distribution similarity for Hurricane. A community challenge (18 teams, 967 agents, 933 evaluated) compares three design paradigms: LLM-as-Brain, LLM-as-Glue, LLM-as-Extra, showing LLM-As-Brain excels on realism in Daily, while LLM-As-Extra performs best on hurricane responsiveness.

**Strengths:**

1. The authors introduce a comprehensive three-dimensional evaluation framework: Robustness, Realism, and Responsiveness, to holistically assess the capability of LLM-based agents in simulating human mobility behavior.
2. The paper presents a large-scale empirical study with over 900 agent deployments, classifying agent design paradigms (LLM as Brain / Glue / Extra) and analyzing performance trade-offs across tasks.
3 The MobiSim-Bench benchmark span both normal (daily) and emergency (hurricane) mobility scenarios. It is grounded in real-world data and includes detailed preprocessing, simulation, and evaluation protocols

**Weaknesses:**

1. The benchmark emphasizes scoring but does not provide a structured way to analyze or debug why an agent fails a particular metric. A qualitative error analysis of LLM-generated anomalies would strengthen the benchmark’s value.
2. Treating robustness mainly as “the program does not crash” is too coarse. The authors should include some more in-depth quantitative measures such as long-horizon agent drift, memory consistency, function-call reliability, and recovery from hallucinated tool outputs. Besides, presenting the variance/standard deviation of all results could further strengthen the paper.
3. The paper lacks discussion on the computational efficiency, latency, or scalability of the presented simulations.
4. Most of the code and data files are missing in the provided anonymous repository. For example, when clicking the files in the baseline and benchmark folders, it shows the message: “The requested file is not found.”

**Questions:**

Please see the weakness section above.

---

### Meta-Review · Area_Chair_W5kA · 2026-01-06

**Summary:**

The submission introduces MobiSim-Bench, an evaluation framework for LLM-agent mobility simulation along robustness, realism, and responsiveness, with two day-level tasks (Daily Mobility using Beijing trajectories with intention annotations, and Hurricane Mobility using Columbia mobility during Hurricane Dorian) and a large-scale challenge evaluation over 18 teams and 967 submitted agents. Reviewers generally value the motivation and scale, but argue that the paper’s positioning and novelty are overstated relative to closely related prior LLM-based mobility simulation work and that the link between the conceptual framework and the concrete benchmark design is not convincingly justified.  Multiple reviewers also highlight substantial missing methodological detail (how agents are instructed, what the rule systems do in the Glue/Extra paradigms, and what exactly constitutes a trajectory), which makes the reported paradigm comparisons difficult to interpret and reproduce. Concerns were further amplified by the coarse robustness definition (essentially pass/fail “does not crash”), lack of computational cost/scalability reporting, limited failure-mode analysis, and reports that key files in the anonymous repository were inaccessible.  After carefully reading the framework description, metric formulas, and the result tables (including the JSD-based Daily Mobility score and the change-rate/distribution-based Hurricane Mobility score), I agree that the benchmark could become useful but that the current manuscript does not yet meet the bar for a benchmark-track contribution due to insufficient grounding and documentation.  Given the mostly negative reviewer consensus and the remaining gaps in clarity, novelty, and reliability, my decision is to recommend rejection at this time.

**Reviewer Concerns:**

The authors did not provide a rebuttal, and therefore none of the reviewers’ concerns were addressed during the rebuttal phase.

**Reviewer Scores:**

The authors did not provide a rebuttal; therefore, none of the reviewers would have been able to revise their scores even if they had participated fully in the discussion.

---

### Decision · Program_Chairs · 2026-01-26

Reject